# CROSS-LINGUAL ABILITY OF MULTILINGUAL BERT: AN EMPIRICAL STUDY

**Karthikeyan K**[*]
Department of Computer Science and Engineering
Indian Institute of Technology Kanpur
Kanpur, Uttar Pradesh 208016, India
kkarthi@cse.iitk.ac.in

**Zihan Wang**[*]
Department of Computer Science
University of Illinois Urbana-Champaign
Urbana, IL 61801, USA
zihanw2@illinois.edu

**Stephen Mayhew**[†]
Duolingo
Pittsburgh, PA, 15206, USA
stephen@duolingo.com

**Dan Roth**
Department of Computer and Information Science
University of Pennsylvania
Philadelphia, PA 19104, USA
danroth@seas.upenn.edu

## ABSTRACT

Recent work has exhibited the surprising cross-lingual abilities of multilingual BERT (M-BERT) – surprising since it is trained without any cross-lingual objective and with no aligned data. In this work, we provide a comprehensive study of the contribution of different components in M-BERT to its cross-lingual ability. We study the impact of linguistic properties of the languages, the architecture of the model, and the learning objectives. The experimental study is done in the context of three typologically different languages – Spanish, Hindi, and Russian – and using two conceptually different NLP tasks, textual entailment and named entity recognition. Among our key conclusions is the fact that the lexical overlap between languages plays a negligible role in the cross-lingual success, while the depth of the network is an integral part of it. All our models and implementations can be found on our project page[1].

## 1 INTRODUCTION

Embeddings of natural language text via unsupervised learning, coupled with sufficient supervised training data, have been ubiquitous in NLP in recent years and have shown success in a wide range of monolingual NLP tasks, mostly in English. Training models for other languages has been shown more difficult, and recent approaches relied on bilingual embeddings that allowed the transfer of supervision in high resource languages like English to models in lower resource languages; however, inducing these bilingual embeddings required some level of supervision (Upadhyay et al., 2016).

Multilingual BERT[2] (M-BERT), a Transformer-based (Vaswani et al., 2017) language model trained on raw Wikipedia text of 104 languages suggests an entirely different approach. Not only is the model contextual, but its training also requires no supervision – no alignment between the languages is done. Nevertheless, and despite being trained with no explicit cross-lingual objective, M-BERT produces a representation that seems to generalize well across languages for a variety of downstream tasks (Wu & Dredze, 2019).

In this work, we attempt to develop an understanding of the success of M-BERT. We study a range of aspects on two different NLP tasks, in order to identify the key components in the success of the model. Our study is done in the context of only two languages, source (typically English) and target (multiple, quite different languages). By involving only a pair of languages, we can study

---

[*]Equal Contribution; most of this work was done while the author interned at the University of Pennsylvania.
[†]This work was done while the author was a student at the University of Pennsylvania.
[1]http://cogcomp.org/page/publication_view/900
[2]https://github.com/google-research/bert/blob/master/multilingual.md

the performance on a given target language, ensuring that it is influenced only by the cross-lingual transfer from the source language, without having to worry about a third language interfering.

We analyze the two-languages version of M-BERT (B-BERT, for bilingual BERT, from now on) in three orthogonal dimensions: (i) linguistic properties and similarities of target and source languages; (ii) network architecture, and (iii) input and learning objective.

One hypothesis that has been discussed as a way to explain the success of M-BERT has to do with some level of language similarity. This could be lexical similarity (shared words or word-parts) or structural similarities (word-ordering or word-frequency), or both. Therefore, we begin by investigating the contribution of *word-piece overlap* – the extent to which the same word-pieces appear in both source and target languages – and distinguish it from other similarities, which we group into a *structural similarity* between the source and target languages. Surprisingly, as we show, B-BERT is cross-lingual even when there is absolutely no word-piece overlap. That is, other aspects of language similarity must be contributing to the cross-lingual capabilities of the model. This is contrary to the Pires et al. (2019) hypothesis that M-BERT gains its power from shared word-pieces. Furthermore, we show that the amount of word-piece overlap in B-BERT's training data contributes little to performance improvements.

Our study of the model architecture addresses the importance of (i) the network depth, (ii) the number of attention heads, and (iii) the total number of model parameters in B-BERT. Our results suggest that depth and the total number of parameters of B-BERT are crucial for both monolingual and cross-lingual performance, whereas multi-head attention is not a significant factor. A single attention head B-BERT can already give satisfactory results.

To understand the role of the learning objective and the input representation, we study the effect of (i) the next sentence prediction objective, (ii) the language identifier in the training data, and (iii) the level of tokenization in the input representation (character, word-piece, or word tokenization). Our results indicate that the next sentence prediction objective actually hurts the performance of the model while identifying the language in the input does not affect B-BERT's performance cross-lingually. Our experiments also show that character-level and word-level tokenization of the input results in significantly worse performance than word-piece level tokenization.

Overall, we provide an extensive set of experiments on three source-target language pairs, English–Spanish, English–Russian, and English–Hindi, selected for variance of script and typological features. We evaluate the performance of B-BERT on two very different downstream tasks: cross-lingual Named Entity Recognition – a sequence prediction task that requires only local context – and cross-lingual Textual Entailment Dagan et al. (2013) that requires more global representation of the text.

Ours is not the first study of M-BERT. Wu & Dredze (2019) and Pires et al. (2019) identified the cross-lingual success of the model and tried to understand it. The former by considering M-BERT layer-wise, relating cross-lingual performance with the amount of shared word-pieces and the latter by considering the model's ability to transfer between languages as a function of word order similarity in languages. However, both works treated M-BERT as a black box, comparing performance on different languages. This work, on the other hand, examines how B-BERT performs cross-lingually by probing its components along multiple aspects.

We also note that some of the architectural conclusions have been observed earlier, if not investigated, in other contexts. Liu et al. (2019) and Yang et al. (2019) argued that the next sentence prediction objective of BERT (the monolingual model) is not very useful; we show that this is the case in the cross-lingual setting. Voita et al. (2019) prunes attention heads for a transformer based machine translation model and argues that most attention heads are not important; in this work, we show that the number of attention heads is not important in the cross-lingual setting.

Our contributions are threefold: (i) we provide the first extensive study of the aspects of M-BERT that give rise to its cross-lingual ability; (ii) we develop a methodology that facilitates the analysis of similarities between languages and their impact on cross-lingual models; we do this by mapping English to a Fake-English language, that is identical in all aspects to English but shares no word-pieces with any target language; finally, (iii) we develop a set of insights into B-BERT, along linguistic, architectural, and learning dimensions, that would contribute to further understanding and to the development of more advanced cross-lingual neural models.

## 2 BACKGROUND

### 2.1 BERT

BERT (Devlin et al., 2019) is a Transformer-based (Vaswani et al., 2017) pre-trained language representation model that has already seen wide use. In contrast to traditional language model objectives that predict the next word given a context, BERT learns to predict the value of masked words (so called, Masked Language Modelling, or MLM (Taylor, 1953)), as well as decide if two sentences are contiguous, known as next sentence prediction (NSP). Input to BERT is a pair of sentences[3] A and B, such that half of the time B comes after A in the original text and the rest of the time B is a randomly sampled sentence. Some tokens from the input are randomly masked, and the MLM objective is to predict the masked tokens. Devlin et al. (2019) argues that MLM enables a deep representation from both directions, and NSP helps understand the relationship between two sentences and can be beneficial to representations.

After BERT is pre-trained on a massive amount of unlabeled text, the representations can then be used in downstream tasks. Typically, a new task-specific layer is added to the top of BERT, and all parameters are fine-tuned on the target task.

### 2.2 MULTILINGUAL BERT

Multilingual BERT is pre-trained in the same way as monolingual BERT except using Wikipedia text from the top 104 languages. To account for the differences in the size of Wikipedia, some languages are sub-sampled, and some are super-sampled using exponential smoothing (Devlin et al., 2018). It's worth mentioning that there are no cross-lingual objectives specifically designed nor any cross-lingual data, e.g. parallel corpus, used.

## 3 WHY MULTILINGUAL BERT WORKS

We present analysis for the cross-lingual ability of multilingual BERT (in our case B-BERT) in three dimensions: (i) linguistic properties and similarities of target and source languages; (ii) network architecture, and (iii) input and learning objective.

### 3.1 DATASETS AND EXPERIMENTAL SETUP

In this work, we conduct all our experiments on two conceptually different downstream tasks – cross-lingual Textual Entailment (TE) and cross-lingual Named Entity Recognition (NER). TE measures natural language understanding (NLU) at a sentence and sentence pair level, whereas NER measures NLU at a token level. We use the Cross-lingual Natural Language Inference (XNLI) (Conneau et al., 2018) dataset to evaluate cross-lingual TE performance and LORELEI dataset (Strassel & Tracey, 2016) for Cross-Lingual NER.

#### 3.1.1 CROSS-LINGUAL NATURAL LANGUAGE INFERENCE (XNLI)

XNLI is a standard cross-lingual textual entailment dataset that extends MultiNLI (Williams et al., 2018) dataset by creating a new dev and test set and manually translating into 14 different languages. Each input consists of a premise and hypothesis pair, and the task is to classify the relationship between premise and hypothesis into one of the three labels: entailment, contradiction, and neutral. While training, both premise and hypothesis are in English, and while testing, both are in the target language. XNLI uses the same set of premises and hypotheses for all the language, making the comparison across languages possible.

#### 3.1.2 CROSS-LINGUAL NAMED ENTITY RECOGNITION (NER)

Named Entity Recognition is the task of identifying and labeling text spans as named entities, such as people's names and locations. The NER dataset (Strassel & Tracey, 2016) we use consists of news and social media text labeled by native speakers following the same guideline in several languages, including English, Hindi, Spanish, and Russian. We subsample 80%, 10%, 10% of English NER

---

[3]In the implementation, the input is a pair of segments, which can contain multiple sentences

data as training, development, and testing. We use the whole dataset of Hindi, Spanish, and Russian for testing purposes. The vocabulary size is fixed at 60000 and is estimated through the unigram language model in the SentencePiece library (Kudo, 2018).

### 3.1.3 NOTATION AND EXPERIMENTAL SETUP

We denote B-BERT trained on language A and B as A-B, e.g., B-BERT trained on English (*en*) and Hindi (*hi*) as *en-hi*, similarly for Spanish (*es*) and Russian (*ru*). For pre-training, we subsample *en*, *es*, and *ru* Wikipedia to 1GB and use the entire Wikipedia for Hindi. Unless otherwise specified, for B-BERT training, we use a batch size of 32, a learning rate of 0.0001, and 2M training steps.

For XNLI, we use the same fine-tuning approach as BERT uses in English and report accuracy. For NER, we extract BERT representations as features and fine-tune a Bi-LSTM CRF model (without character-wise embeddings) and report $F_1$ score averaged from 5 runs and its standard deviation.

### 3.2 LINGUISTIC PROPERTIES

Pires et al. (2019) hypothesizes that the cross-lingual ability of M-BERT arises because of the shared word-pieces between source and target languages. However, our experiments show that B-BERT is cross-lingual even when there is no word-piece overlap. Similarly, Wu & Dredze (2019) hypothesizes that source language should be selected such that it shares more word-pieces with the target language while our experiment suggests that structural similarity (e.g. word-ordering) is much more important. Motivated by the above hypotheses, in this section, we study the contribution of word-piece overlap and structural similarity for the cross-lingual ability of B-BERT.

### 3.2.1 WORD-PIECE OVERLAP

M-BERT is trained using Wikipedia text from 104 languages, and the texts from different languages share some common word-piece vocabulary (like numbers, links, etc.. including actual words, if they have the same script), we refer to this as word-piece overlap. Previous work (Pires et al., 2019) hypothesizes that M-BERT generalizes across languages because these shared word-pieces force the other word-pieces to be mapped to the same shared space.

In this section, we perform experiments to compare cross-lingual performance with and without word-piece overlap. We construct a new corpus – Fake-English (*enfake*), by shifting the Unicode of each character in English Wikipedia text by a large constant so that there is strictly no character overlap with any other Wikipedia text [4]. We may consider Fake-English as a different language than English, but having the exact same properties except word surface forms.

We measure the contribution of word-piece overlap as the drop in performance when the word-piece overlap is removed. From Table 1, we can see B-BERT *is cross-lingual even when there is no word-piece overlap*. We can also see that the contribution of word-piece overlap is very small, which is quite surprising and contradictory to prior hypotheses (Pires et al., 2019; Wu & Dredze, 2019).

### 3.2.2 WORD-ORDERING SIMILARITY

Words are ordered differently between languages. For example, English has a Subject-Verb-Object order, while Hindi has a Subject-Object-Verb order. We analyze whether similarity in how words are ordered affects learning cross-lingual transferability. We study the effect of word-ordering similarity by destroying the word-ordering structure through randomly permuting some percentage of words in sentences during pre-training. We permute both source (Fake-English) and target language (although permuting any one of them would also be sufficient). This way, the similarity of word-ordering is hidden from B-BERT. We quantify the amount of permutation by sampling $25\%, 50\%, 100\%$ of the $\binom{L}{2}$ pairs of word-pieces in a sentence with $L$ word-pieces, and swap each pair (e.g. $wp_i, ..., wp_j$ becomes $wp_j, ..., wp_i$). This way of shuffling by no means generates a uniform random distribution on permutations, but provides a roughly good way to control the randomness. Note that for each

---

[4]Before feeding the text to BERT's transformer model, there is an additional vocabulary embedding step, so that each word-piece gets mapped to a certain integer (unrelated to its unicode). Therefore, BERT will not have access to the exact values of the unicode, and will not be able to learn directly from the linear shift of unicode we introduced.

| B-BERT | Train | Test | XNLI | | NER |
| --- | --- | --- | --- | --- | --- |
| | | | Accuracy | Word-piece Contribution | F$_1$-Score |
| en-es | en | es | 72.3 | 1.4 | 61.9 (±0.8) |
| enfake-es | enfake | | 70.9 | | 62.6 (±1.6) |
| en-hi | en | hi | 60.1 | 0.5 | 61.6 (±0.7) |
| enfake-hi | enfake | | 59.6 | | 62.9 (±0.7) |
| en-ru | en | ru | 66.4 | 0.7 | 57.1* (±0.9) |
| enfake-ru | enfake | | 65.7 | | 54.2 (±0.7) |
| en-enfake | enfake | enfake | 78.0 | 0.5 | 78.9* (±0.7) |
| en-enfake | enfake | en | 77.5 | | 76.6 (±0.8) |

Table 1: **The Effect of Word-piece Overlap and of Structural Similarity** For different pairs of B-BERT languages, and for two tasks (XNLI and NER), we show the contribution of word-pieces to the success of the model. In every two consecutive rows, we show results for a pair (e.g., English-Spanish) and then for the corresponding pair after mapping English to a disjoint set of word-pieces. The gap between the performance in each group of two rows indicates the loss due to completely eliminating the word-piece contribution. We add an asterisk to the number for NER when the results are statistically significant at the 0.05 level.

word-piece, the other word-pieces that appear in its context (other word-pieces in the sentence) are not changed, although the order is changed. We also do not permute during fine-tuning, as we only want to control cross-lingual ability gained during pre-training.

| B-BERT | Permutation amount | XNLI (acc) | NER (F$_1$-Score) |
| --- | --- | --- | --- |
| Pre-train: enfake-es | 0.0 | 70.9 | 62.6 (±1.6) |
| Fine-tune: en | 0.25 | 68.9 | 42.3 (± 0.6) |
| Evaluate: es | 0.5 | 65.5 | 41.3 (±3.8) |
| | 1.0 | 62.5 | 39.4 (±1.5) |
| Pre-train: enfake-hi | 0.0 | 59.6 | 62.9 (±0.7) |
| Fine-tune: en | 0.25 | 51.4 | 24.8 (±2.0) |
| Evaluate: hi | 0.5 | 48.3 | 8.7 (±1.9) |
| | 1.0 | 43.1 | 2.9 (±1.9) |
| Pre-train: enfake-ru | 0.0 | 65.7 | 54.2 (±0.7) |
| Fine-tune: en | 0.25 | 63.6 | 23.6 (±2.3) |
| Evaluate: ru | 0.5 | 59.7 | 16.3 (±0.4) |
| | 1.0 | 53.6 | 16.3 (±0.9) |

Table 2: **Contribution of Word-Ordering similarity:** We study the importance of word-order similarity by analysing the performance of XNLI and NER when some percent of word-order similarity is reduced. The percent $p$ controls the amount of similarity (how random each sentence is permuted). We can see that word-order similarity is quite important, however there must be other components of structural similarity that could contribute for the cross-lingual ability, as the performance of almost random is still passable.

From Table 2, we can see that the performance drops significantly when we curtail the word-order similarity between the two languages. However, the cross-lingual performance is still significantly better than random, which indicates that there are other components of structural similarity, which could contribute to the cross-lingual ability of B-BERT.

### 3.2.3 WORD-FREQUENCY SIMILARITY

We also study whether only knowing unigram word frequency allows for good cross-lingual representations. Indeed, Zipf's law indicates that words appear with different frequency, and one may suggest similar meaning words appear with relatively similar frequency in a pair of languages, helping B-BERT learn cross-lingually. We collect the frequency of words in the target language and generate a new monolingual corpus by sampling words based on the frequency, i.e., each sentence is a set of random words sampled from the original unigram frequency. The only information B-BERT learns from the languages is their unigram frequency, and perhaps sub-word level information and distribution of lengths of sentences. We train B-BERT using Fake-English and this newly generated target corpus. From Table 3, we can see that the performance is very poor. Therefore, unigram frequencies alone don't contain enough information for cross-lingual learning.

| B-BERT | Is Frequency Based | XNLI (acc) | NER ($F_1$-Score) |
|---|---|---|---|
| enfake-es | No | 70.9 | 62.6 ($\pm$1.6) |
|  | Yes | 35.4 | 15.0($\pm$ 1.1) |
| enfake-hi | No | 59.6 | 62.9 ($\pm$0.7) |
|  | Yes | 36.3 | 1.2 ($\pm$0.9) |
| enfake-ru | No | 65.7 | 54.2 ($\pm$0.7) |
|  | Yes | 34.4 | 5.7 ($\pm$0.5) |

Table 3: **Cross-lingual ability from only unigram frequency:** We study if only unigram frequency is useful for cross-lingual transfer. enfake-es indicates B-BERT trained with Fake-English and the new created corpus, where each sentence is a set of random words sampled from the same unigram distribution as es. The results show that only unigram frequency is not enough for a reasonable cross-lingual performance.

### 3.2.4 STRUCTURAL SIMILARITY

We define the structure of a language as every property of an individual language that is invariant to the script of the language (e.g., morphology, word-ordering, word frequency are all parts of structure of a language). From Table 1, we can see that BERT transfers very well from Fake-English to English. Also note that, despite not sharing any vocabulary, Fake-English transfers to Spanish, Hindi, Russian almost as well as English. On XNLI, where the scores between languages can be compared, the cross-lingual transferability from Fake-English to Spanish is much better than from Fake-English to Hindi/Russian. Since they do not share any word-pieces, this better transferability comes from the structure similarity being closer between Spanish and Fake-English.

Through Section 3.2.2 we know that between two languages, even when there are no word-piece overlap, nor any particular order of words in a sentences, B-BERT still learns some cross-lingual features. Section 3.2.3 shows from the other side that if B-BERT is given only little information (unigram frequency), it learns almost no cross-lingual features. These results suggest that we should shed more light on studying the structural similarity between languages, for example, higher order of $k-$gram or $k-$co-occurrence frequencies (notice that the word ordering experiment can be seen as giving B-BERT $k-$ co-occurrence word frequencies where $k$ is the length of the longest sentence in the corpus). In this study, we don't further dissect the structure of language. Despite its amorphous definition, our experiment clearly shows that structural similarity is crucial for cross-lingual transfer.

### 3.3 MODEL ARCHITECTURE

From Section 3.2, we observe that B-BERT recognizes language structure effectively. In this section, we hypothesize this ability is an emergent property of the model architecture. We study the contribution of different components of B-BERT architecture namely (i) depth, (ii) multi-head attention and (iii) the total number of parameters. The motivation is to understand which components are crucial for its cross-lingual ability.

We perform all our cross-lingual experiments on the XNLI dataset with Fake-English as the source and Russian as the target language; we measure cross-lingual ability by the difference between the performance of Fake-English and Russian (lesser the difference better the cross-lingual ability).

### 3.3.1 DEPTH

We presume the ability of B-BERT to extract good semantic and structural features is a crucial reason for its cross-lingual effectiveness, and the deepness of B-BERT helps it extract good language features. In this section, we study the effect of depth on both the monolingual and cross-lingual performance of B-BERT. We fix the number of attention heads and change the size of hidden units and intermediate units such that the total number of parameters are almost the same (size of intermediate units is always $4\times$ size of hidden units).

From Table 4, we can see that deeper models not only perform better on English but are also better cross-lingually. We can also see a strong correlation between performance on English and cross-lingual ability ($\Delta$), which further supports our assumption that the ability to extract good semantic and structural features is a crucial reason for its cross-lingual effectiveness.

| Parameters (in Millions) | Depth | Multi-head Attention | XNLI | | |
|---|---|---|---|---|---|
| | | | Fake-English | Russian | Δ |
| 138.69 | 1 | 12 | 66.6 | 45.0 | 21.6 |
| 136.32 | 2 | 12 | 73.7 | 55.7 | 18.0 |
| 136.20 | 4 | 12 | 76.9 | 59.0 | 17.9 |
| 138.86 | 6 | 12 | 78.3 | 63.1 | 15.2 |
| 136.10 | 18 | 12 | 79.1 | 66.0 | 13.1 |
| 139.33 | 24 | 12 | 78.9 | 67.6 | 11.3 |
| 132.78 | 12 | 12 | 79.0 | 65.7 | 13.3 |

Table 4: **The Effect of Depth of B-BERT Architecture:** We use Fake-English and Russian B-BERT and study the effect of depth of B-BERT towards XNLI. We vary the depth and fix both the number of attention heads and the number of parameters – the size of hidden and intermediate units are changed so that the total number of parameters remains almost the same. We train only on Fake-English and test on both Fake-English and Russian and report their test accuracy. The difference between the performance on Fake-English and Russian(Δ) is our measure of cross-lingual ability (lesser the difference, better the cross-lingual ability).

### 3.3.2 MULTI-HEAD ATTENTION

In this section, we study the effect of multi-head attention on the cross-lingual ability of B-BERT. We fix the depth and the total number of parameters – which is a function of depth and size of hidden and intermediate layers and study the performance for the different number of attention heads. From Table 5, we can see that the number of attention heads doesn't have a significant effect on cross-lingual ability (Δ). B-BERT is satisfactorily cross-lingual even with a single attention head, which is in agreement with the recent study on monolingual BERT (Voita et al., 2019; Clark et al., 2019).

| Parameters (in Millions) | Depth | Multi-head Attention | XNLI | | |
|---|---|---|---|---|---|
| | | | Fake-English | Russian | Δ |
| 132.78 | 12 | 1 | 77.4 | 63.2 | 14.2 |
| 132.78 | 12 | 2 | 78.3 | 62.8 | 15.5 |
| 132.78 | 12 | 3 | 79.5 | 65.3 | 14.2 |
| 132.78 | 12 | 6 | 78.9 | 66.7 | 12.2 |
| 132.78 | 12 | 16 | 77.9 | 64.9 | 13.0 |
| 132.78 | 12 | 24 | 77.9 | 63.9 | 14.0 |
| 132.78 | 12 | 12 | 79.0 | 65.7 | 13.3 |

Table 5: **The Effect of Multi-head Attention:** We study the effect of the number of attention heads of B-BERT on the performance of Fake-English and Russian language on XNLI data. We fix both the number of depth and number of parameters of B-BERT and vary the number of attention heads. The difference between the performance on Fake-English and Russian (Δ) is our measure of cross-lingual ability.

### 3.3.3 TOTAL NUMBER OF PARAMETERS

Similar to depth, we also anticipate that a large number of parameters could potentially help B-BERT extract good semantic and structural features. We study the effect of the total number of parameters on cross-lingual performance by fixing the number of attention heads and depth; we change the number of parameters by changing the size of hidden and intermediate units (size of intermediate units is always $4\times$ size of hidden units). From Table 6, we can see that the total number of parameters is not as significant as depth; however, below a threshold, the number of parameters seems significant, which suggests that B-BERT requires a certain minimum number parameters to extract good semantic and structural features.

### 3.3.4 GENERALIZATION TO BERT WITH MORE LANGUAGES

Here we show that the results for model structure hold also for a more multilingual case; to further illustrate this, we experiment on four language BERT (en, es, hi, ru). From Table 7, we can see that the performance on XNLI is comparable even with just 15% of parameters, and just 1 or 3 attention heads when the depth is good enough, which is in agreement with our observations in the main text.

### 3.4 INPUT AND LEARNING OBJECTIVE

In this section, we study the effect of input representation and learning objectives on cross-lingual ability of B-BERT. Recall that BERT is trained with Masked Language Modeling (MLM) and Next Sentence Prediction (NSP) objectives. We study the effect of NSP as recent works (Conneau & Lample, 2019; Joshi et al., 2019; Liu et al., 2019) show that the NSP objective hurts the performance on several monolingual tasks. To verify whether B-BERT can learn in language agnostic settings, we also study the use of language identity markers. We are also interested in studying the effect of tokenization and language representation, using characters and words vocabulary instead of word-pieces.

### 3.4.1 NEXT SENTENCE PREDICTION (NSP)

The input to BERT is a pair of sentences separated by a special token such that half the time the second sentence is the next and rest half the time, it is a random sentence. The NSP objective of BERT (B-BERT) is to predict whether the second sentence comes after the first one in the original text. We study the effect of NSP objective by comparing the performance of B-BERT pre-trained with and without this objective. From Table 8, we can see that the NSP objective hurts the cross-lingual performance even more than monolingual performance.

| Parameters (in Millions) | Depth | Multi-head Attention | XNLI | | |
|---|---|---|---|---|---|
| | | | Fake-English | Russian | $\Delta$ |
| 7.87 | 3 | 3 | 68.5 | 43.2 | 25.3 |
| 12.19 | 3 | 3 | 70.1 | 44.1 | 26.0 |
| 16.78 | 3 | 3 | 70.8 | 50.4 | 20.4 |
| 8.40 | 6 | 6 | 70.2 | 49.7 | 20.5 |
| 13.37 | 6 | 6 | 72.4 | 56.2 | 16.2 |
| 18.87 | 6 | 6 | 73.3 | 54.4 | 18.9 |
| 4.23 | 12 | 12 | 65.5 | 46.6 | 18.9 |
| 11.83 | 12 | 12 | 71.3 | 56.3 | 15.0 |
| 29.65 | 12 | 12 | 76.6 | 61.4 | 15.2 |
| 132.78 | 12 | 12 | 79.0 | 65.7 | 13.3 |
| 283.11 | 12 | 12 | 79.6 | 65.4 | 14.2 |

Table 6: **The Effect of Total Number of Parameters:** We study the effect of the total number of parameters of B-BERT on the performance of Fake-English and Russian language on XNLI data. We fix both the number of depth and number of attention heads of B-BERT and vary the total number of parameters by changing the size of hidden and intermediate units. The difference between the performance on Fake-English and Russian ($\Delta$) is our measure of cross-lingual ability.

| Parameters (in Millions) | Depth | Multi-head Attention | XNLI | | | |
|---|---|---|---|---|---|---|
| | | | en | es | hi | ru |
| 132.78 (100%) | 12 | 12 | 79.0 | 70.0 | 50.7 | 65.2 |
| 20.05 (15.1%) | 16 | 1 | 74.1 | 63.2 | 49.6 | 58.9 |
| 20.05 (15.1%) | 16 | 3 | 75.0 | 65.3 | 51.1 | 60.3 |
| 24.20 (18.23%) | 24 | 1 | 75.0 | 64.5 | 48.3 | 58.6 |
| 24.20 (18.23%) | 24 | 3 | 75.7 | 65.5 | 48.9 | 60.2 |
| 30.43 (22.92%) | 36 | 1 | 75.2 | 66.5 | 47.1 | 60.3 |
| 30.43 (22.92%) | 36 | 3 | 76.0 | 64.8 | 47.8 | 59.8 |

Table 7: **Similar results on M-BERT with four languages:** We show that the insights derived from bilingual BERT is also valid in the case of multilingual BERT (4 language BERT). Further, we also show that with enough depth, we only need a fewer number of parameters and attention heads to get comparable results.

| B-BERT | Train | Test | XNLI | | NER | |
|---|---|---|---|---|---|---|
| | | | NSP | No-NSP | NSP | No-NSP |
| enfake-es | enfake | enfake es | 78.5 70.9 | 78.7 72.7 | 80.3 (±0.6) 62.6 (±1.6) | 80.7 (±1.4) 64.6 (±1.4) |
| enfake-hi | enfake | enfake hi | 79.3 59.6 | 80.1 60.7 | 81.4 (±0.9) 62.9 (±0.7) | 80.0 (±1.1) 62.4 (±1.4) |
| enfake-ru | enfake | enfake ru | 79.0 65.7 | 79.0 66.7 | 80.2 (±0.7) 54.2 (±0.7) | 80.3 (±0.8) 55.7 (±0.3) |

Table 8: **Effect of Next Sentence Prediction Objective:** We study the effect of NSP objective on XNLI and NER. Column NSP and No-NSP show the performance (accuracy for XNLI and average (stdev) F1-score for NER) when B-BERT is trained with and without NSP objective respectively. The difference between the NSP and No-NSP shows that NSP objective hurts performance.

### 3.4.2 LANGUAGE IDENTITY MARKER

In Section 3.2, we argue that B-BERT is cross-lingual because of its ability to recognize language structure similarities, and hence we presume adding a language identity marker doesn't affect its cross-lingual ability. Even if we don't add a language identity marker, BERT learns language identity (Wu & Dredze, 2019). To incorporate language identity in the input, we add different end of string tokens ([SEP]) for different languages (i.e., our input format is [CLS] SENT1 [SEP-A] SENT2 [SEP-B], where A and B are languages corresponding to SENT1 and SENT2 respectively). From Table 9, we can see that adding language identity marker doesn't affect cross-lingual performance.

| B-BERT | Train | Test | XNLI | | NER | |
|---|---|---|---|---|---|---|
| | | | No Lang-id | With Lang-id | No Lang-id | With Lang-id |
| enfake-es | enfake | enfake es | 78.5 70.9 | 78.4 72.2 | 80.3 (±0.6) 62.6(±1.6) | 81.7 (±1.1) 62.2 (±0.4) |
| enfake-hi | enfake | enfake hi | 79.3 59.6 | 79.0 59.6 | 81.4 (±0.9) 62.9 (±0.7) | 80.7 (±1.6) 61.0 (±0.7) |
| enfake-ru | enfake | enfake ru | 79.0 65.7 | 78.4 65.3 | 80.2 (±0.7) 54.2 (±0.7) | 79.1 (±1.8) 55.7 (±0.6) |

Table 9: **Effect of Language Identity Marker in the Input:** We study the effect of adding a language identifier in the input data. We use different end of string ([SEP]) tokens for different languages serving as language identity marker. Column "With Lang-id" and "No Lang-id" show the performance when B-BERT is trained with and without language identity marker in the input.

### 3.4.3 CHARACTER VS. WORD-PIECE VS. WORD

We compare the performance of B-BERT with character, word-piece, and word tokenized input. For character B-BERT, we use all the characters as vocabulary, and for word B-BERT, we use the most frequent 100000 words as vocabulary. From Table 10, we can see that the cross-lingual performance (difference between source and target language performance) of B-BERT with word-piece tokenized is similar to that of word tokenized, while both are better than character tokenized. We believe that this is because word-pieces and words carry much more information than characters, allowing B-BERT to learn similarities between the two languages easier.

## 4 DISCUSSION AND FUTURE WORK

This paper provides a systematic empirical study addressing the cross-lingual ability of B-BERT. The analysis covers three dimensions: (i) linguistic properties and similarities of target and source languages; (ii) network architecture, and (iii) input and learning objective.

In order to gauge the language similarity aspect needed to make B-BERT successful, we created a new language – Fake-English – and this allows us to study the effect of word-piece overlap while maintaining all other properties of the source language. Our experiments reveal some interesting and

| B-BERT | Train | Test | XNLI | | | NER | | |
|--------|-------|------|------|-----------|------|------|-----------|------|
| | | | Char | WordPiece | Word | Char | WordPiece | Word |
| enfake-es | enfake | enfake | 73.7 | 80.0 | 80.3 | 78.8 (±1.3) | 80.3 (±1.5) | 74.9 (± 2.2) |
| | | es | 66.6 | 74.9 | 74.4 | 62.0 (±0.8) | 64.8 (±0.9) | 57.5 (±0.4) |
| enfake-hi | enfake | enfake | 73.9 | 80.3 | 80.0 | 79.6 (±0.9) | 79.7 (±1.1) | 75.0 (± 1.9) |
| | | hi | 53.8 | 61.7 | 60.3 | 53.1 (±0.4) | 58.8 (±1.2) | 56.6 (±0.8) |
| enfake-ru | enfake | enfake | 74.2 | 80.7 | 79.2 | 77.2 (±1.1) | 80.8 (±1.3) | 73.8 (± 0.9) |
| | | ru | 61.4 | 68.1 | 65.0 | 52.1 (±0.5) | 56.5 (±0.3) | 46.4 (±1.3) |

Table 10: **Effect of Character vs Word-Piece vs Word tokenization**. We compare the performance of B-BERT with different tokenized input on XNLI and NER data. Column *Char, Word-Piece, Word* reports the performance of B-BERT with character, word-piece and work tokenized input respectively. We use 2k batch size and 500k epochs.

surprising results. The most notable finding is that word-piece overlap on the one hand, and multi-head attention on the other, are both not significant, whereas structural similarity and the depth of B-BERT are crucial for its cross-lingual ability.

While in order to better control interference among languages, we studied the cross-lingual ability of B-BERT instead of those of M-BERT, it would be interesting now to extend this study, allowing for more interactions among languages. We leave it to future work to study these interactions. In particular, one important question is to understand the extent to which adding to M-BERT languages that are *related* to the target language, helps the model's cross-lingual ability.

We introduced the term *Structural Similarity*, despite its obscure definition, and show its significance in cross-lingual ability. Another interesting future work could be to develop a better definition and, consequently, a finer set of experiments, to better understand the *Structural similarity* and study its individual components.

Finally, we note an interesting observation made in Table 11. We observe a drastic drop in the entailment performance of B-BERT when the premise and hypothesis are in different languages. (This data was created using XNLI where in the original form the languages contain the same premise and hypothesis pair). One of the possible explanations could be that BERT is learning to make textual entailment decisions by matching words or phrases in the premise to those in the hypothesis. This question, too, is left as a future direction.

| B-BERT | Target | Premise Language – Hypothesis Language (XNLI) | | | |
|--------|--------|-------------|-------------|-------------|-------------|
| | | enfake-target | target-enfake | enfake-enfake | target-target |
| enfake-es | es | **57.9** | **61.1** | 78.5 | 70.9 |
| enfake-hi | hi | **45.7** | **55.6** | 79.3 | 59.6 |
| enfake-ru | ru | **51.1** | **57.9** | 79.0 | 65.7 |

Table 11: **Premise and Hypothesis in different language:** Using XNLI test set, we construct textual entailment data with premise and hypothesis in different languages. The column A-B (e.g. enfake-target) refers to test data with premise in language A (enfake) and hypothesis in language B (target). We always train on Fake-English and report test accuracy.

ACKNOWLEDGMENTS

This work was supported by Contracts W911NF-15-1-0461, HR0011-15-C-0113, and HR0011-18-2-0052 from the US Defense Advanced Research Projects Agency (DARPA), by Google Cloud, and by Cloud TPUs from Googles TensorFlow Research Cloud (TFRC). The views expressed are those of the authors and do not reflect the official policy or position of the Department of Defense or the U.S. Government or Google.

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
