# OpenReview forum: "Cross-Lingual Ability of Multilingual BERT: An Empirical Study"
_ICLR.cc/2020/Conference — Accept (Poster)_

### Official Review · AnonReviewer2 · 2019-10-24
**Official Blind Review #2**

**Rating:** 3

**Review:**


CONTRIBUTIONS:

C1. Cross-linguistic token overlap. Fake-English: English, but with the Unicode codes of English characters all shifted by a large constant so that there is no overlap between Fake-English characters and those of actual languages, but the language-internal structure remains that of English.

C2. A bilingually-trained BERT, pretrained on languages L and L’, is then trained on a downstream task in L then tested on that task in L’. The task is Cross-Lingual NLI (XNLI) or Cross-Lingual NER. L’ is Spanish, Hindi, or Russian. L is English or Fake-English. Comparing the success at test when L = English vs. when L = Fake-English, it is shown that eliminating all token overlap between L and L’ has a small effect (less than 1.5% on XNLI, less than about 3.5% on NER). (Table 1)

C3. Several architectural parameters of BERT are varied holding the others roughly constant (same tasks as C2, with L = Fake-English). This shows that depth (Table 2) and level of tokenization matter (Table 7), while little effect results from varying the number of attention heads (Table 3), number of parameters (Table 4), whether the next sentence prediction task is used for training (Table 5), or whether the language of an input is explicitly given (Table 6).

C4. Testing cross-language entailment on XNLI by B-BERT shows that there is a large reduction in performance when the hypothesis and premises sentences are from different languages.

RATING: Weak reject

REASONS FOR RATING (SUMMARY). Aside from the invention of Fake-English, which as far as I know is original and a clever approach to assessing the importance of token overlap in cross-language transfer, the other contributions are reporting results of mechanical changes. The paper’s contributions are useful, but do not reach a level of generality, originality, or depth justifying presentation at ICLR.

Although it did not factor into my rating, I would like to point out that saying ‘structural similarity is important’ and saying ‘word-piece overlap is not important’ is saying exactly the same thing twice, since the gain not attributable to word-piece overlap, by their definition, equals the gain due to ‘structural similarity’, which is a concept otherwise undefined and unmeasurable.

**Experience Assessment:**

I have read many papers in this area.

**Review Assessment: Checking Correctness Of Derivations And Theory:**

N/A

**Review Assessment: Checking Correctness Of Experiments:**

I carefully checked the experiments.

**Review Assessment: Thoroughness In Paper Reading:**

I read the paper thoroughly.

---

> ### Comment · AnonReviewer2 · 2019-11-15
> **AnonReviewer2 Response**
>
> Having read the other reviews, I feel even more strongly in the summary of the reasons given for the rating in my original review. I am tempted to lower my score, but have decided to keep it at 3: Weak Reject.

---

> ### Author Response · Authors · 2019-11-15
> **Significance of our work**
>
> We sincerely thank the reviewer for reviewing our paper.
> > **Lack of generality, originality or depth**
>
> (1) First, we would like to point out that this paper is the first to propose an experimental design that proves that word-pieces overlap do not contribute to the transferability of M-BERT. This was done by inventing the notion of Fake-English, with a distinct word-piece space. Moreover, we believe that the methodology we propose in this paper is general enough to support additional insights, and will be followed up by other authors, and therefore this in itself is a significant contribution.
>
> (2) Second, while the design of our architectural experiments may not be sophisticated, we are the first to perform this set of experiments systematically, and identify the aspects of the architecture that are important for transferability, as well as those that are not. We believe that this, too, is an important contribution to understanding M-BERT. Further, we have added a few more results on the number of parameters to understand the threshold, and also showed that we can get comparable performance with only a small number of parameters and attention heads, even in multilingual case (four language BERT). Please refer to appendix section “FURTHER DISCUSSIONS ON ARCHITECTURE”
>
> (3) Third, our results are the first to show clearly that the transferability of M-BERT depend on some aspect of structural similarity between the languages, and has nothing to do with lexical similarity. While we have not isolated yet which aspects of structural similarity contribute to transferability, and how much, this is already an important contribution, please refer to the appendix section “FURTHER  DISCUSSIONS ON STRUCTURAL SIMILARITY”, for some of our initial experiments that break this down a bit more.
>
> (4) Our final observation of the drastic drop in performance when the premise and hypothesis are in different languages (Table 8) might suggest that BERT is simply learning the word matching instead of learning the actual entailment. This observation definitely needs special attention to understand what BERT learns from entailment supervision.
>
>
> General comment:
> Also, please take a look at the general comments for more on structural similarity and the number of parameters experiment.

---

### Official Review · AnonReviewer3 · 2019-10-24
**Official Blind Review #3**

**Rating:** 6

**Review:**

This paper evaluates the cross-lingual effectiveness of Multilingual BERT along three dimensions:
- Linguistics
- Architecture
- Input and learning objective

In each of these three dimensions, the authors run experiments to test why BERT is effective at cross-lingual transfer.
For simplicity, they run their experiments on B-BERT which is trained on two languages. The fine-tuning is done on language, and the zero-shot performance is tested on the other one.

Pros:
I found the en-fake experiments enlightening. The authors find that wordpiece overlap is not as important for cross-lingual transfer as was suggested by previous papers (Wu & Pires).
The idea of creating a unicode shifted version of English and use it for testing is a first of its kind and quite interesting.
Most experiments were well motivated and the authors draw good conclusions about the need for more depth, that only a few attention heads are sufficient.
They end with an experiment that shows that the cross-lingual effectiveness drops significantly when the premise and hypothesis are in different languages. This is a good motivating experiment to end the paper on.

Cons:
- The architecture experiments were not that insightful and they authors did not reach concrete suggestions on a minimum number of parameters or a minimum depth.
- While the two language setting is easier to experiment with, I wonder how these conclusions will change if they were repeated with the 100+ language version.
- For structural similarity, it would have been more concrete if the authors were able to visualize and show that the newly created en-fake still aligned with corresponding similar words in the other languages. This would have proven that despite no wordpiece overlap, similar words still align.


Minor comments:
- typo: "training also training also"
- bad grammar - last para in page 1
- I found the introduction was filled with grammar and bad English. Please fix.
- Why are the numbers in Table 4 in a different format?
- 3.4.2 It's not clear if this is a clear trend. The authors claim that lang-id helps.
- Please explicitly state the sentencepiece or wordpiece setup in a central piece. I found the detail hidden in section 3.1.2
- With the word vs char vs wordpiece experiments, I think more care should be taken to make sure that the number of parameters remains the same across all three setups. e.g. with only a few chars as vocab, the model has far fewer parameters. This should be compensated for.

**Experience Assessment:**

I have published one or two papers in this area.

**Review Assessment: Checking Correctness Of Derivations And Theory:**

I carefully checked the derivations and theory.

**Review Assessment: Checking Correctness Of Experiments:**

I carefully checked the experiments.

**Review Assessment: Thoroughness In Paper Reading:**

I read the paper thoroughly.

---

> ### Author Response · Authors · 2019-11-15
> **More insights on Architecture and Multilingual settings**
>
> We sincerely thank the reviewer for reviewing our paper.
>
> >**The architecture experiments did not reach concrete suggestions on a minimum number of parameters or a minimum depth.**
>
> -- We agree that we didn’t show the concrete minimum number of parameters, but now we added few more results for number of parameters, and we think the trend is clear now (There is a drastic drop in performance when the number of parameters is changed from 11.83M to 7.23M, this is kind of threshold, at least for 12 layer and 12 attention settings). Please refer to appendix section “FURTHER DISCUSSIONS ON ARCHITECTURE”
>
> -- We think the trend with depth is mostly clear
> For English: The performance is almost saturated after a depth of 6
> For Russian (for cross-lingual): It’s almost saturated from 12 (still the performance increases slightly)
> Also, the performance of English drops slightly when we go from 18 to 24 layers. So, around this range is quite good for cross-lingual transfer.
>
> >** How these conclusions will change if they were repeated with the 100+ language version.**
>
> -- We didn’t do the 100+ language version but we currently added in the appendix the 4 language version (we hope the results follows similarly even for 100+ version)
> --  Our results further show that we can get comparable performance even with as little as 15% of parameters, and single attention, given that the depth is good enough.
>
> General comment:
> Also, please take a look at the general comments for more on structural similarity and the number of parameters experiment.

---

### Official Review · AnonReviewer1 · 2019-10-29
**Official Blind Review #1**

**Rating:** 6

**Review:**

What is the task?
Comprehensive study of the contribution of different components in Multilingual BERT to its cross-lingual ability.

What has been done before?
(Wu & Dredze, 2019) and (Pires et al., 2019) identified the cross-lingual success of the model and tried to understand it. However, both works treated M-BERT as a black box and compared M-BERT’s performance on different languages. This work, on the other hand, examines how B-BERT performs cross-lingually by probing its components, along multiple aspects. Some of the architectural conclusions have been observed earlier, if not investigated, in other contexts.

Authors claim that “Pires et al. (2019) hypothesizes that the cross-lingual ability of M-BERT arises because of the shared word-pieces between source and target languages.” is not entirely correct. Pires et al. (2019) did show M-BERT’s ability to transfer between languages that are written in different scripts, and thus have effectively zero lexical overlap. There were results (e.g. Figure 1) showing that while performance using EN-BERT depends directly on word piece overlap, M-BERT’s performance is largely independent of overlap, indicating that it learns multilingual representations deeper than simple vocabulary memorization.

Pires et al. (2019) also showed that structural similarity is crucial for cross-lingual transfer

What are the main contributions of the paper?
First comprehensive study of the contribution of different components in Multilingual BERT to its cross-lingual ability. Novel findings about the effect of network architecture, input representation and learning objective on cross lingual ability of M-BERT
Methodology that facilitates the analysis of similarities between languages and their impact on cross-lingual models by mapping English to a Fake-English language, that is identical in all aspects to English but shares no word-pieces with any target language.

What are the key dimensions studied/analyzed?
Different components/aspects of Multilingual BERT investigated:
(i) Linguistics properties and similarities of target and source languages  (has been studied in prior work)
(ii) Network Architecture (novel)
(iii) Input and Learning Objective (moderately novel)

What are the main results? Are they significant?
Lexical overlap between languages plays a negligible role in the cross-lingual success, while the depth of the network is an important part of it.

Strengths
Novel findings about the effect of network architecture, input representation and learning objective on cross lingual ability of M-BERT
Weaknesses
Pires et al. (2019) work has been misrepresented. (see above for more details)
Pires et al. (2019) did  study linguistics properties and similarities of target and source languages for Multilingual BERT and had similar findings as this work.


Questions
What kind of difference in the numbers is considered significant by the authors ? For example, according to them, an increase of more than 2 points in accuracy in Table 3 (e.g. 1 head vs. 6 heads) is considered insignificant. But a decrease of a point in accuracy in Table 5  (e.g. enfake-ru NSP vs. No-NSP) is significant.


**Experience Assessment:**

I have published one or two papers in this area.

**Review Assessment: Checking Correctness Of Derivations And Theory:**

N/A

**Review Assessment: Checking Correctness Of Experiments:**

I carefully checked the experiments.

**Review Assessment: Thoroughness In Paper Reading:**

I read the paper at least twice and used my best judgement in assessing the paper.

---

> ### Author Response · Authors · 2019-11-15
> **Clarification: Pires et al. and significant difference**
>
> We sincerely thank the reviewer for valuable comments.
>
> > **Wrong interpretation of Pires et al. (2019)**
>
> -- We agree that Pires et al. also showed that M-BERT transfers between languages written in different scripts. However, they reason that this cross-lingual ability comes from the small number of word-piece overlap beyond lexical, such as numbers and URLs (Section 6 in [1]).
>
> Indeed an excerpt from Pires et al.: “As to why M-BERT generalizes across languages, we hypothesize that having word pieces used in all languages (numbers, URLs, etc) which have to be mapped to a shared space forces the co-occurring pieces to also be mapped to a shared space, thus spreading the effect to other word pieces, until different languages are close to a shared space”
>
> A key contribution of the paper is that we are the first to suggest a solid experimental design that proves that word-piece overlap is not the reason for the transferability supported by M-BERT.
>
>
>
> > **What kind of difference in the numbers is considered significant by the authors ?**
>
> --  In the case of the number of attention heads, significance means "what is its importance for cross-lingual transfer as a whole (or what is its importance in comparison to other components of architecture)".  By saying insignificant, we are not suggesting to use single attention, but arguing that it does not affect cross-lingually much. Similarly, in the case of word-piece overlap, we just say that it is not a major factor for cross-lingual transferability, but in terms of absolute performance, we still lose about 1%.
> -- Whereas, in Next Sentence Prediction, significance means the difference comes beyond randomness, such that it is advisable to remove the NSP objective (we already know that with NSP it is cross-lingual)/
>
> General Comment:
> -- Also, please take a look at the general comments for more on structural similarity.
>
> [1] Pires, Telmo, Eva Schlinger, and Dan Garrette. "How multilingual is Multilingual BERT?." arXiv preprint arXiv:1906.01502 (2019).

---

### Author Response · Authors · 2019-11-15
**General Comment (for all 3 reviewers) -- Structural Similarity and Architecture**

We have updated our paper with additional experiments that strengthen our contributions. Please refer to the appendix.

Structural Similarity:
We found that reviewers and others are slightly concerned about the structural similarity, mainly due to its abstract nature. To illustrate the necessity of structural similarity and make it lucid, we have added an analysis of 2 sub-components of structural similarity:

(1) Effect of Word-ordering
            (a) Words are ordered differently between languages. For example, English has a Subject-Verb-Object order, while Hindi has a Subject-Object-Verb order. We analyze this component of structural similarity.
            (b) We destroyed the word-ordering -- one component of structural similarity --  by shuffling some percentage of the words in sentences during pretraining. We shuffle both the source (Fake-English) and the target language (shuffling any one of them would also be sufficient).  This way, the word ordering component of the structure is hidden from B-BERT.  We shuffle random 25%, 50% and 100% of the words in the sentence while keeping others in their respective positions. When the sentence is 100% shuffled, each sentence can be treated as a Bag of Words.
            (c) Note that during fine-tuning we don’t permute -- as cross-lingual ability arises from the pretraining, and not from fine-tuning.
            (d) Our conclusion is that word ordering is crucial, but cross-linguality still preserves even when the whole sentence is shuffled. Please refer to A.1.1 WORD-ORDERING SIMILARITY for more details.


(2) Effect of word-frequencies (frequency distribution)
            (a) It is possible that good cross-lingual representations benefit from similar words in languages having a similar frequency. In the perfect similar language (English-Fake), the same words have exactly the same frequency.
            (b) Here, we study whether only word frequency (unigram frequency) allows for good cross-lingual representations
            (c) We collect the frequency of words in the target language and generate a new monolingual corpus where a sentence is a set of words sampled from the same unigram frequency distribution as the original target language.
            (d) Our conclusion is that when BERT is only given the frequency of words of the target language, the cross-lingual ability is very poor, but surprisingly not trivial.  Please refer to A.1.2 WORD-FREQUENCY SIMILARITY for more details.


Architecture:

As reviewer 3 asked for some concrete threshold of number of parameters, we added a few more results on the number of parameters experiments, and we think the trend is clear now (There is a drastic drop in performance when the number of parameters is changed from 11.83M to 7.23M, this is kind of threshold, at least for 12 layer and 12 attention settings).

Please refer to appendix section “FURTHER DISCUSSIONS ON ARCHITECTURE”

---

### Decision · Program_Chairs · 2019-12-19

**Decision:**

Accept (Poster)

**Comment:**

This paper introduces a set of new analysis methods to try to better understand the reasons that multilingual BERT succeeds. The findings substantially bolster the hypothesis behind the original multilingual BERT work: that this kind of model discovers and uses substantial structural and semantic correspondences between languages in a fully unsupervised setting. This is a remarkable result with serious implications for representation learning work more broadly.

All three reviewers saw ways in which the paper could be expanded or improved, and one reviewer argued that the novelty and scope of the paper are below the standard for ICLR. However, I am inclined to side with the two more confident reviewers and argue for acceptance. I don't see any substantive reasons to reject the paper, the methods are novel and appropriate (even in light of the prior work that exists on this question), and the results are surprising and relevant a high-profile ongoing discussion in the literature on representation learning for language.